# Post-operative patient-related risk factors for chronic pain after total knee replacement: a systematic review

Vikki Wylde, Andrew D Beswick, Jane Dennis, Rachael Gooberman-Hill

Musculoskeletal Research Unit, Bristol Medical School, University of Bristol, Bristol, UK

**Correspondence to**
Dr Vikki Wylde;
v.wylde@bristol.ac.uk

## ABSTRACT

**Objective** To identify postoperative patient-related risk factors for chronic pain after total knee replacement (TKR).

**Design** The systematic review protocol was registered on the International Prospective Register of Systematic Reviews (CRD42016041374). MEDLINE, Embase and PsycINFO were searched from inception to October 2016 with no language restrictions. Key articles were also tracked in the Institute for Scientific Information (ISI) Web of Science. Cohort studies evaluating the association between patient-related factors in the first 3 months postoperatively and pain at 6 months or longer after primary TKR surgery were included. Screening, data extraction and assessment of methodological quality were undertaken by two reviewers. The primary outcome was pain severity in the replaced knee measured with a patient-reported outcome measure at 6 months or longer after TKR. Secondary outcomes included adverse events and other aspects of pain recommended by the core outcome set for chronic pain after TKR.

**Results** After removal of duplicates, 16 430 articles were screened, of which 805 were considered potentially relevant. After detailed evaluation of full-text articles, 14 studies with data from 1168 participants were included. Postoperative patient-related factors included acute pain (eight studies), function (five studies) and psychosocial factors (four studies). The included studies had diverse methods for assessment of potential risk factors and outcomes, and therefore narrative synthesis was conducted. For all postoperative factors, there was insufficient evidence to draw firm conclusions about the association with chronic pain after TKR. Selection bias was a potential risk for all studies, as none were reported to be conducted at multiple centres.

**Conclusion** This systematic review found insufficient evidence to draw firm conclusions about the association between any postoperative patient-related factors and chronic pain after TKR. Further high-quality research is required to provide a robust evidence base on postoperative risk factors, and inform the development and evaluation of targeted interventions to optimise patients' outcomes after TKR.

## INTRODUCTION

Primary total knee replacement (TKR) is a common operation, with over 100 000 operations performed in the UK in 2015,[1 2] and demand is projected to increase dramatically.[3]

### Strengths and limitations of this study

► This is the first systematic review of patient-related risk factors for chronic pain after total knee replacement.
► Meta-analysis was not possible due to heterogeneity in the assessment of risk factors and outcomes.
► We did not include studies that used a composite pain and function measure to assess outcome.

Patients choose to have a TKR to relieve chronic pain and improve functional ability,[4] but approximately 20% of patients experience chronic postsurgical pain,[5 6] defined as pain present at 3 months after surgery.[7] The impact of chronic pain after TKR is considerable and patients may struggle to cope and adjust to this pain.[8] Provision of services for patients with chronic pain after TKR is patchy and inconsistent,[9] with a lack of explicit access points.[10] A systematic review identified that only one intervention has been evaluated for the management of this condition: a single intra-articular botulinum toxin injection.[11]

The identification of risk factors for chronic pain after TKR is a fundamental step in designing interventions to improve patient outcomes. Understanding the relevance of non-modifiable factors, such as sex and ethnicity, can help patients and clinicians work together to make informed decisions about TKR. Although some factors may not be modifiable, others may be amenable to intervention. Identification of modifiable patient-related risk factors is an important element in the development of interventions to improve outcomes after TKR. Previous systematic reviews have synthesised the literature on preoperative risk factors for chronic pain after TKR.[12–15] These reviews have found evidence for a range of modifiable preoperative patient-related risk factors, including pain intensity, catastrophising, mental health and comorbidities. Preoperative interventions have largely focused on exercise and

education and have shown little long-term postoperative benefit.[15] Further interventions specifically targeting pain-related behaviours, such as cognitive-behavioural patient education and pain-coping skills training, are being evaluated.[16 17]

While the potential value for preoperatively identifying at-risk patients and targeting them with appropriate interventions is clear, multivariable models have been found to have low predictive power, explaining less than 10% of the variability in chronic pain.[18] An operation itself is an important risk factor for chronic pain,[19] and factors relating to the operation and early recovery may be important risk factors. A risk index including presurgical variables and acute postsurgical pain had 'fair' predictive power for the development of chronic postsurgical pain across diverse surgery types.[20] Therefore, in addition to evaluating preoperative risk factors, it is important to consider postsurgical factors that may limit rehabilitation and recovery, and be associated with chronic pain. If patients at risk of developing chronic pain could be identified in the early postoperative period, targeted interventions could be delivered, potentially as part of a comprehensive perioperative care package, to prevent the development of chronic pain. Although trials evaluating the effectiveness of early postoperative interventions on reducing chronic pain have been conducted,[21–24] no systematic review has yet evaluated postoperative risk factors for chronic pain after TKR. Therefore, the aim of this systematic review was to identify early postoperative patient-related risk factors for chronic pain after TKR.

## METHODS
### Protocol and registration
The protocol was registered on the International Prospective Register of Systematic Reviews (PROSPERO) on 6 July 2016 (reference: CRD42016041374). Conduct and reporting of this systematic review adhere to recommendations from the Preferred Reporting Items for Systematic Reviews and Meta-Analyses[25] (online supplementary appendix 1).

### Eligibility criteria
Studies were eligible for inclusion in the review if they met the following criteria:

### Population
Adults undergoing primary TKR predominantly for osteoarthritis: Studies that included patients with TKR combined with patients undergoing other orthopaedic procedures were included if separate results were available for patients with TKR.

### Exposure
Postoperative patient-related risk factors measured in the first 3 months after surgery: Patients with exposure were those with a risk factor (categorical variable) or higher level of risk factor (continuous variable). The focus of this review was on patient-related risk factors with the potential for modification or use in targeting care, and therefore studies that assessed clinical risk factors (eg, length of stay, postoperative complications or radiographical measurements) or analgesic use were excluded.

### Comparator
Patients with absence of risk factor (categorical variable) or lower level of risk factor (continuous variable).

### Outcome
Severity of pain in the replaced knee measured with a patient-reported outcome measure at 6 months or longer after TKR surgery.

### Study design
Cohort studies that have explored the relationships between factors measured in the first 3 months postoperative and longer term pain outcomes.

### Information sources and searches
MEDLINE, Embase and PsycINFO were searched from inception to 17 October 2016. Searches were conducted by experienced systematic reviewers (AB and JD) based on established design filters.[26 27] The search strategy combined terms relating to study design (eg, cohort, epidemiological study) and population (eg, knee replacement, knee arthroplasty). Full search strategies are provided in online supplementary appendix 2. No language restrictions were applied. Searches were supplemented with hand-searching of reference lists and review articles, and key articles were tracked in the ISI Web of Science. Conference abstracts were excluded. ClinicalTrials.gov was searched on 18 August 2017 for ongoing observational studies and records screened in duplicate by two reviewers (JD and VW).

### Study selection and data extraction
Bibliographical details of the articles identified were exported into EndNote X7 (Thomson Reuters) and duplicates removed. After an initial screening of titles and abstracts by one reviewer (AB) to remove clearly irrelevant studies, titles and abstracts were screened in duplicate by two reviewers (AB and VW). As recommended in the Cochrane Handbook,[28] reviewers were 'over inclusive' at early stages and retained any potentially relevant studies. Full texts of all such reports were acquired and assessed for eligibility against the PICOS criteria in duplicate by two reviewers (AB and VW). Discrepancies were resolved in discussion with a third reviewer (JD). Data from articles that met the eligibility criteria were extracted into an Excel database by one reviewer (VW), with checking against source articles by a second reviewer (AB or JD). Extracted data comprised country, date, setting, population, participant demographics, study methodology including statistical analysis, risk factors, time to follow-up, losses to follow-up, joint-specific pain outcomes, variables included in multivariable analyses, and information relevant to assessment of study methodological quality.

Where necessary, authors of studies were contacted for further information to enable judgements about eligibility and/or to provide unpublished outcome data relevant to the review. If data from patients with TKR were combined with patients undergoing other orthopaedic procedures, separate data for patients with TKR were requested. If a combined pain and function outcome was reported, such as the Oxford Knee Score or Western Ontario and McMaster Universities Osteoarthritis Index (WOMAC) score, separate pain-specific data were requested, for example, the Oxford Knee Score pain subscale or WOMAC Pain Scale.

## Outcomes

The primary outcome was pain severity in the replaced knee measured with a patient-reported outcome measure at 6 months or longer after TKR. Chronic postsurgical pain is defined as pain present at 3 months after surgery[7]; however, research has shown that most of the improvement in pain occurs in the first 3–6 months after TKR surgery.[29–32] Therefore, 6 months postoperative was deemed an appropriate time point to assess chronic pain. Secondary outcomes included adverse events and other aspects of pain recommended by the core outcome set for chronic pain after TKR.[33] These included pain interference with daily living, pain and physical functioning, temporal aspects of pain, pain description, emotional aspects of pain, use of pain medication and satisfaction with pain relief. No limits were placed on the tools used to measure these outcomes.

## Assessment of methodological quality of included studies

The Newcastle-Ottawa Quality Assessment Scale[34] and ROBINS-I tool[35] are established tools for the assessment of risk of bias in randomised controlled trials and studies reporting non-randomised controlled comparisons. However, risk of bias assessment in systematic reviews of observational studies is less well established. The MINORs tool[36] has been developed; however, this is a summative checklist, and as such risks rating reporting rather than conduct.[37] Therefore we developed a non-summative checklist for use in this review. This checklist consisted of four items to assess selection bias (inclusion of consecutive patients and representativeness), bias due to missing data (follow-up rates) and bias due to inadequate consideration of confounding (multivariable or univariable analysis). These items were informed by existing tools, including the MINORs, Newcastle-Ottawa Quality Assessment Scale and the ROBINS-I tool. Each item was rated as adequate, not adequate or not reported. Each individual item rating is reported, rather than an overall score. Ratings of methodological quality for included studies were conducted independently by two reviewers (VW and JD), and any discrepancies were resolved in discussion with a third reviewer (AB).

## Data synthesis

In the protocol, meta-analyses were planned if two or more studies assessed the same risk factor with suitable methodology. In comparing groups of patients with or without a risk factor, outcomes adjusted for baseline patient factors would be considered in preference to unadjusted outcomes, and the effect of non-adjustment would be explored in a subgroup analysis. If studies reported categorical pain outcomes, risk ratios would be used to summarise cohort studies and ORs for case–control studies. For risk factors reported as continuous variables, results of meta-analyses would be reported as mean differences or standardised mean differences, depending on the consistency of risk factor and outcome measures reported. We planned to explore the effect of non-adjustment for other variables in a subgroup analysis. Assessment of heterogeneity was planned using the $\chi^2$ and $I^2$ statistic. The protocol stated that we would conduct sensitivity analyses on methodological quality assessment.

At analysis stage, opportunities for meta-analysis were limited by heterogeneity in the assessment of risk factors and outcomes. Therefore, we undertook a descriptive narrative analysis, in keeping with the approach recommended by the Cochrane Handbook.[28]

## RESULTS

After removal of duplicates, 16 430 articles were screened, of which 857 were considered potentially relevant. After detailed evaluation of full-text articles, 14 studies with data from 1613 participants were included[38–51] (figure 1). The most common reasons for excluding potentially relevant studies were because patient-related factors were not assessed and follow-up after TKR surgery was less than 6 months.

### Study characteristics

An overview of study characteristics is provided in table 1.

Of the 14 included studies, three were from the UK, two each from Australia, USA and Spain, and one study from Belgium, Denmark, France, Portugal and Serbia. Thirteen studies were conducted at a single centre and one study did not report the number of centres. Eleven of the studies were cohort studies, two were randomised controlled trials retrospectively analysed as cohort studies and one was a case–control study with prospective data collection. Sample sizes ranged from 23 to 402, with a median of 115 participants. One study included a small number of patients undergoing unicompartmental knee replacement but was included in the review as 83% of participants had TKR.[49] Follow-up assessments varied: four studies assessed outcomes at 6 months after TKR, five at 12 months and the remainder between 3 and 7 years postoperatively. Pain at follow-up was evaluated using the WOMAC Pain Scale[52] (five studies), Numerical Rating Scale (NRS; three studies), Visual Analogue Scale (VAS; two studies), American Knee Society Score pain question[53] (two studies), and Verbal Descriptor Scale (VDS; two studies). Secondary outcomes for the review relating to serious adverse events and other aspects of

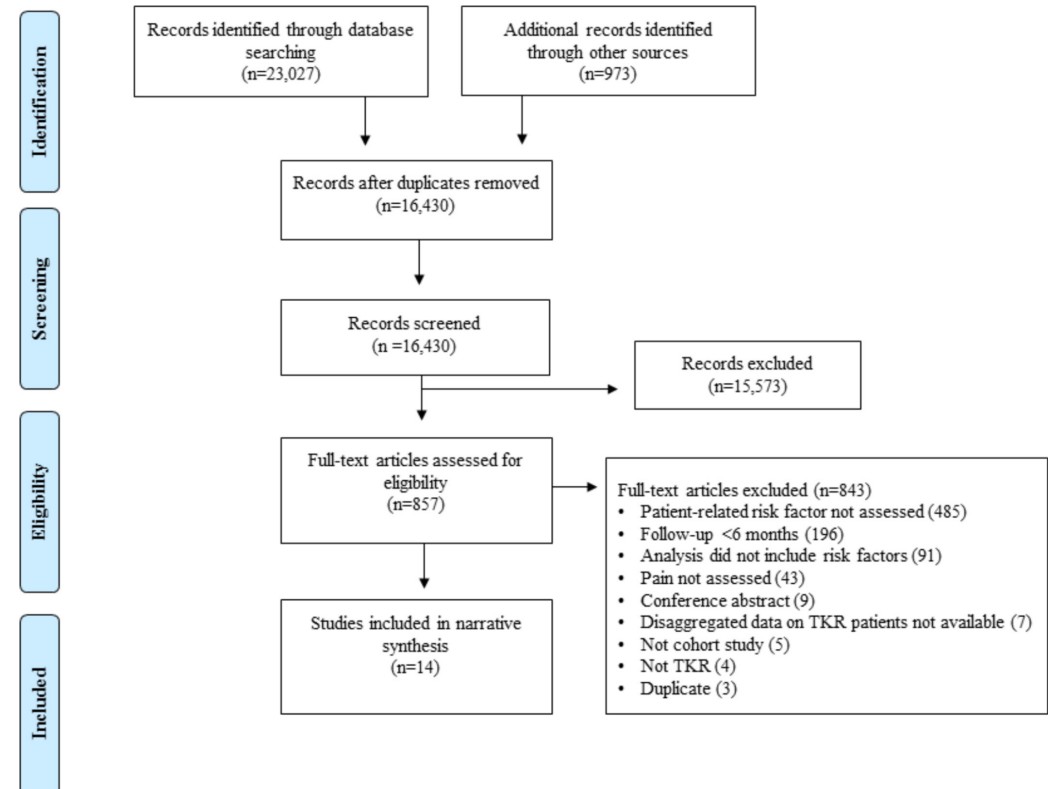

**Figure 1** Systematic review flow diagram. TKR, total knee replacement.

**Table 1** Characteristics of included studies

| Study | Dates of baseline data collection | Study design | Country | Participants recruited/at final follow-up | Mean/ median age | Female (%) | Outcome measure | Duration of follow-up |
|---|---|---|---|---|---|---|---|---|
| Crosbie et al[38] | 2005–2006 | Cohort* | Australia | 102/100 | 68 | 56 | WOMAC pain | 6 months |
| Edwards et al[39] | Not reported | Cohort | USA | 43 in analysis | 72 | 58 | VAS | 12 months |
| Elson and Brenkel[40] | 1995–1998 | Case–control | UK | 622/402 knees | 69 | 54 | AKSS pain question | 5 years |
| Grosu et al[41] | 2009–2010 | Cohort | Belgium | 114/68 | 66 | 66 | VDS | 12 months |
| Núñez et al[42] | 2000–2001 | Cohort | Spain | 88/67 | 75 | 81 | WOMAC pain | 3 years |
| Núñez et al[43] | 2000 | Cohort | Spain | 142/112 | 67 | 77 | WOMAC pain | 7 years |
| Phillips et al[44] | 2009–2010 | Cohort | UK | 96/80 | 71 | 56 | VAS | 39–51 months |
| Pinto et al[45] | 2009–2011 | Cohort | Portugal | 42 in analysis | 66 | 77 | NRS | 4–6 months |
| Riis et al[46] | 2007–2009 | Cohort | Denmark | 176/154 | 68 | 65 | AKSS pain question | 12 months |
| Sayers et al[47] | 2009–2012 | Cohort* | UK | 316/277 | 69 | 53 | WOMAC pain | 12 months |
| Stephens et al[48] | Not reported | Cohort | USA | 71/63 | 67 | 54 | WOMAC pain | 6 months |
| Thomazeau et al[49] | 2013 | Cohort | France | 109/104 | 69 | 72 | NRS | 6 months |
| Kocic et al[50] | 2007–2013 | Cohort | Serbia | 78/78 | 68 | 76 | NRS | 6 months |
| Veal et al[51] | 2013 | Cohort | Australia | 23 in analysis | Not available | Not available | VDS | 12 months |

*Retrospective analysis of randomised controlled trial data.
AKSS, American Knee Society Score; NRS, Numerical Rating Scale; VAS, Visual Analogue Scale; VDS, Verbal Descriptor Scale; WOMAC, Western Ontario and McMaster Universities Osteoarthritis Index.

**Table 2** Ratings of methodological quality for included studies

| Study | Inclusion of consecutive patients | Representativeness (multicentre adequate) | Percentage follow-up (>80% adequate) | Minimisation of potential confounding (multivariable analysis adequate) |
|---|---|---|---|---|
| Crosbie et al[38] | + | – | + | + |
| Edwards et al[39] | | – | – | + |
| Elson and Brenkel[40] | | – | – | – |
| Grosu et al[41] | | – | – | – |
| Núñez et al[42] | + | – | – | + |
| Núñez et al[43] | + | – | – | + |
| Phillips et al[44] | + | – | + | – |
| Pinto et al*[45] | + | – | – | + |
| Riis et al[46] | + | – | + | + |
| Sayers et al[47] | +† | – | + | + |
| Stephens et al[48] | | | + | + |
| Thomazeau et al[49] | + | – | + | + |
| Kocic et al[50] | | – | + | – |
| Veal et al*[51] | | – | + | – |

*For studies where authors provided data on patients with total knee replacement, ratings are based on the study as reported in the article.
†Information obtained through personal contact.
+, adequate; –, inadequate; blank, not reported.

pain outcomes were infrequently reported and therefore not summarised.

### Assessment of methodological quality of included studies

Ratings of methodological quality for the 14 included studies are provided in table 2. Eight studies reported that consecutive patients were recruited, eight studies followed up >80% participants, and nine studies conducted multivariable analysis. All studies had issues relating to selection bias because none were reported as being conducted at multiple centres.

### Patient-related postoperative risk factors

Patient-related postoperative risk factors were categorised into three groups: acute postoperative knee pain, knee function and psychosocial factors.

### Acute postoperative knee pain

Eight studies including data from 737 participants evaluated the association between pain in the first 3 months after TKR and chronic pain (table 3). Timing of acute postoperative pain was classified as pain within the first postoperative week; pain between 1 and 2 weeks postoperatively; and pain from 2 weeks to 3 months. Pain as a risk factor was assessed using the VAS (three studies), VDS (two studies), NRS (two studies), WOMAC Pain Scale (one study) and PainDETECT (one study). Five studies conducted multivariable analysis, two studies conducted univariable analysis, and for one study no statistical analysis was performed as data were provided by authors on a small subset of patients with TKR.

### Pain severity on postoperative days 1–7

Four studies with data from 491 participants evaluated whether pain severity in the first week after surgery was associated with chronic pain.[41 45 47 49] Two were at risk of bias due to missing data and one study was at risk of bias due to inadequate consideration of confounding. Methods used to assess pain included the VDS,[41] VAS[47] and NRS.[45 49] Three studies found that more severe acute postoperative pain was associated with more severe pain at 6–12 months after TKR,[41 47 49] although in one study this association was attenuated completely after adjustment for preoperative pain.[47] One study found no association between pain at 42 hours after surgery and the presence of chronic pain at 4–6 months.[45]

### Pain severity in postoperative days 8–14

Three studies with data from 191 participants evaluated whether pain severity on postoperative days 8–14 was associated with chronic pain.[38 41 51] One study was at risk of bias due to missing data and two studies were at risk of bias due to inadequate consideration of confounding. Pain was assessed in two studies with the VDS[41 51] and in one with the WOMAC Pain Scale and VAS.[38] Pain on postoperative day 8 and at 2 weeks was not found to be associated with chronic pain in two studies,[38 41] and descriptive data only were available for the study that evaluated pain on postoperative day 10.[51] In the study with low risk of bias apart from with regard to representativeness,[38] pain severity at 2 weeks was not found to be associated with pain at 6 months after TKR.

**Table 3** Studies evaluating acute postoperative knee pain as a risk factor for chronic pain after TKR

| Author and year | Number in analysis | Risk factor measurement | Outcome(s) | Univariable or multivariable analysis | Association | Results summary |
|---|---|---|---|---|---|---|
| Edwards et al, 2009[39] | 43 | Global pain VAS at 1 month and 3 months | Global pain VAS at 6 and 12 months | Multivariable generalised estimating equation model | Yes | Global pain at a previous time point was a predictor of global pain at a future time point (estimate=0.43, SE=0.08, t=5.8, p<0.001). |
| | | Night pain VAS at 1 month and 3 months | Night pain VAS at 6 and 12 months | | Yes | Night pain at a previous time point was a predictor of night pain at a future time point (estimate=0.32, SE=0.08, t=3.8, p<0.001). |
| Crosbie et al, 2010[38] | 100 | WOMAC Pain Scale at 2 weeks | WOMAC Pain Scale at 6 months | Multivariable linear regression | No | Not significant, results not reported |
| | | VAS at 2 weeks | | | No | Not significant, results not reported |
| | | WOMAC pain at 8 weeks | | | Yes | Beta coefficient=+0.25±0.07 |
| | | VAS at 8 weeks | | | No | Not significant, results not reported |
| Pinto et al, 2013[45] | 42 | NRS at 48 hours | NRS at 4–6 months | Hierarchical logistic regression | No | Exp(B)=0.998 (95% CI 0.623 to 1.601), p=0.995 |
| Phillips et al, 2014[44] | 80 | PainDETECT at 6 weeks | Pain VAS at 39–51 months | Univariable correlation | Yes | PainDETECT at 6 weeks correlated moderately with VAS pain scores (r=0.53). |
| Veal et al, 2015[51] | 23 | VDS for average pain at 10 days | VDS for average pain at 12 months | N/A—statistical analysis inappropriate as data provided by authors on a small subset of patients | N/A | 11 patients had none/mild pain at 10 days, none of these patients had severe/moderate pain at 12 months. 12 patients had moderate/severe pain at 10 days, 2 of these patients had moderate/severe at 12 months. |
| | | VDS for worst pain at 10 days | VDS for worse pain at 12 months | | | 2 patients had none/mild pain at 10 days, none of these patients had severe/moderate pain at 12 months. 21 patients had moderate/severe pain at 10 days, 8 of these patients had moderate/severe at 12 months. |
| | | VDS for average pain at 6 week | VDS for average pain at 12 months | | | 17 patients had none/mild pain at 6 weeks, 1 of these patients had moderate/severe pain at 12 months. 6 patients had moderate/severe pain at 6 weeks, 1 of these patients had moderate/severe at 12 months. |

Continued

**Table 3** Continued

| Author and year | Number in analysis | Risk factor measurement | Outcome(s) | Univariable or multivariable analysis | Association | Results summary |
|---|---|---|---|---|---|---|
| | | VDS for worst pain at 6 weeks | VDS for worse pain at 12 months | | | 9 patients had none/mild pain at 6 weeks, 1 of these patients had severe/moderate pain at 12 months. 14 patients had moderate/severe pain at 6 weeks, 7 of these patients had moderate/severe at 12 months. |
| Grosu et al, 2016[41] | 68 | VDS on days 1, 2 and 3 (cumulative value of maximal pain intensity) | VDS at 6 months VDS at 12 months | Univariable correlation | Yes Yes | r=0.350, p=0.009 r=0.350, p=0.009 |
| | | VDS on day 8 | VDS at 6 months VDS at 12 months | | No No | Not significant, results not reported Not significant, results not reported |
| | | VDS on day 30 | VDS at 6 months VDS at 12 months | | Yes No | r=0.310, p=0.013 Not significant, results not reported |
| | | VDS at 3 months | VDS at 6 months VDS at 12 months | | No No | Not significant, results not reported Not significant, results not reported |
| Sayers et al, 2016[47] | 277 | VAS for pain on rest on days 1, 2 and 3 (combined) | WOMAC pain at 12 months | Multivariable structural equation modelling | Yes | Beta=0.222, SE=0.058, 95% CI 0.109 to 0.336, p=0.0001 When preoperative pain added: beta=0.09, 95% CI −0.09 to 0.27, p=0.332 |
| | | VAS for pain on movement on days 1, 2 and 3 (combined) | | | Yes | Beta=0.140, SE=0.044, 95% CI 0.054 to 0.226, p=0.0014 When preoperative pain added: beta=0.00, 95% CI −0.14 to 0.15, p=0.955 |
| Thomazeau et al, 2016[49] | 104 | NRS on days 1–4 | NRS at 6 months | Multivariate logistic regression | Yes | Patients with high-intensity acute postoperative pain (defined through latent class growth analysis) were more likely to have pain at 6 months than patients with low-intensity acute postoperative pain (OR=4.23, 95% CI 1.39 to 12.88, p=0.011). |

N/A, not applicable; NRS, Numerical Rating Scale; TKR, total knee replacement; VAS, Visual Analogue Scale; VDS, Verbal Descriptor Scale; WOMAC, Western Ontario and McMaster Universities Osteoarthritis Index.

### Pain severity between 2 weeks and 3 months postoperatively

Five studies with data from 314 participants evaluated whether pain severity between 2 weeks and 3 months postoperatively was associated with chronic pain after TKR.[38 39 41 44 51] Two studies were at risk of bias due to missing data and three studies were at risk of bias due to inadequate consideration of confounding. Methods to assess pain were the WOMAC Pain Scale,[38] VAS[38 39 44]

and VDS.[41 51] In one study with risk of bias associated only with conduct at a single centre, pain severity at 8 weeks postoperatively was found to be associated with pain at 6 months postoperatively when assessed with the WOMAC but not the VAS.[38] In one study with univariable analysis, pain severity assessed on day 30 was found to be associated with pain severity at 6 months but not 12 months after TKR.[41] The same study found that pain at 3 months postoperatively was not associated with pain severity at 6 months and 12 months postoperatively.[41] In another study, neuropathic pain at 6 weeks postoperatively was found to be moderately associated with pain at 39–51 months after surgery.[44] In one study, there was no difference in pain at 12 months in patients with different average pain levels at 6 weeks.[51] However considering 'worst' pain, 7/14 patients with moderate to severe pain at 6 weeks reported moderate to severe pain at 12 months compared with 1/9 patients with none or mild pain at 6 weeks. A study that assessed global pain and night pain at 1 month and 3 months postoperatively found that they were associated with global pain and night pain, respectively, at a future time point (6 months and 12 months).[39]

## Knee function

Five studies including data from 835 participants evaluated the association between postoperative knee function and chronic pain after TKR (table 4). Three studies were at risk of bias due to missing data and one study was at risk of bias due to inadequate consideration of confounding. Assessment of knee function varied and included range of motion, ambulatory status, WOMAC function, 6 min walk test and stair ascent speed.

Four studies including data from 735 participants evaluated whether function at hospital discharge was associated with chronic pain after TKR.[40 42 43 46] Two of these studies assessed range of motion[40 46] and two assessed ambulatory status at discharge[42 43]; none found an association. One study, at low risk of bias except inclusion of a single centre, with 100 patients evaluated whether function at 2 weeks and 8 weeks, assessed using three different methods, was associated with WOMAC pain scores at 6 months postoperatively.[38] This study found that WOMAC function score at 2 weeks, but not 8 weeks, was associated with chronic pain; 6 min walk test at both 2 weeks and 8 weeks was associated with chronic pain; stair ascent speed at 2 and 8 weeks was not associated with chronic pain.

**Table 4** Studies evaluating postoperative knee function as a risk factor for chronic pain after TKR

| Author and year | Number in analysis | Risk factor measurement | Outcome | Univariable or multivariable analysis | Association | Results summary |
|---|---|---|---|---|---|---|
| Elson and Brenkel, 2006[40] | 402 knees | Range of motion (active and passive) at hospital discharge | AKSS pain at 5 years | Univariable analysis | No | Not significant, results not reported |
| Núñez et al, 2007[42] | 67 | Ambulatory status at hospital discharge | WOMAC pain at 3 years | Multivariable linear regression | No | Not significant, results not reported |
| Núñez et al, 2009[43] | 112 | Ambulatory status at hospital discharge | WOMAC pain at 7 years | Multivariable linear regression | No | Not significant, results not reported |
| Crosbie et al, 2010[38] | 100 | WOMAC function at 2 weeks | WOMAC pain at 6 months | Multivariable linear regression | Yes | Beta coefficient=+0.06, SE=±0.02 |
| | | 6 min walk test at 2 weeks | | | Yes | Beta coefficient=−0.05, SE=±0.01 |
| | | Stair ascent speed at 2 weeks | | | No | Not significant, results not reported |
| | | WOMAC function at 8 weeks | | | No | Not significant, results not reported |
| | | 6 min walk test at 8 weeks | | | Yes | Beta coefficient=−0.04, SE=±0.01 |
| | | Stair ascent speed at 8 weeks | | | No | Not significant, results not reported |
| Riis et al, 2014[46] | 154 | Range of flexion (active) at hospital discharge | AKSS pain at 12 months | Multivariable binary logistic regression | No | OR 1.00 (95% CI 0.99 to 1.04), p=0.698 |

AKSS, American Knee Society Score; TKR, total knee replacement; WOMAC, Western Ontario and McMaster Universities Osteoarthritis Index.

## Psychosocial factors

Four studies including data from 226 participants evaluated the association between postoperative psychological factors and chronic pain after TKR (table 5). Two studies were at risk of bias due to missing data and one study was at risk of bias due to inadequate consideration of confounding. Risk factors assessed included catastrophising, depression, social support, coping skills, fear of movement and anxiety. In one study, catastrophising at a previous time point was a risk factor for night pain, but not global pain, at a future time point.[39] In the same study, depression was found to be a risk factor for global pain but not night pain. Another study assessing risk factors at 6 weeks postoperatively found that perceived positive social support was associated with less chronic pain, negative social support with more chronic pain, and no association between coping and pain at 6 months after TKR.[48] Patients with a high fear of movement at 2 weeks postoperatively reported more pain at 6 months than those with a low fear of movement.[50] Greater anxiety at 48 hours after surgery was found to be associated with a higher risk of having a pain score of >3 on the NRS at 4–6 months after TKR.[45]

## Ongoing studies

Searches of ClinicalTrials.gov identified five ongoing studies that are collecting data on patient-related postoperative risk factors and pain outcomes at 6 months or longer after TKR. An overview of these studies is provided in online supplementary appendix 3.

## DISCUSSION

This is the first systematic review to evaluate postoperative patient-related risk factors for chronic pain after TKR. Fourteen cohort studies were identified which evaluated the association between patient-related factors measured in the first 3 months postoperatively and pain severity measured with a patient-reported outcome measure at 6 months or longer after primary TKR. Postoperative factors assessed included pain (eight studies), function (five studies) and psychosocial factors (four studies).

For all postoperative patient-related factors, there was insufficient evidence to draw firm conclusions on the association with chronic pain after TKR. When reviewing observational cohort studies, it is essential to consider issues that may introduce bias and lead to potentially misleading results and their interpretation. The key issues relate to generalisability, incomplete follow-up and accounting for confounding factors. Regarding generalisability, findings from single-centre and multicentre studies can differ,[54] and one potential factor contributing to this difference is the recruitment of a more homogeneous population in single-centre studies. The population may be highly selected and therefore have limited validity external to the study setting. Losses to follow-up represent another cause of bias as patients who do not complete longer term assessments may have poorer

outcomes.[55 56] In this review, six studies had data on <80% participants at follow-up. The methodological quality of five studies was limited by the lack of multivariable analysis to minimise the impact of potential confounding on results. In studies with no risk of bias other than patient selection, there was a suggestion that chronic pain was associated with increased acute postoperative pain during the hospital stay.[47 49] However, in one of these studies, a comprehensive assessment of pain relationships over time suggested that the association was largely explained by preoperative pain.[47] For later pain assessments, one study did not identify consistent associations between postoperative pain and chronic pain.[38]

This review has strengths and weaknesses that should be considered when interpreting the results. While our search terms were broad to identify cohort studies that involved patients with TKR, three studies were identified through methods other than the main searches. This is a recognised issue in the identification of observational studies[57] and highlights the importance in bibliographical databases of appropriate indexing and use of keywords. It is possible that studies including general orthopaedic or surgical populations may have included patients with TKR, and these may not have been identified. However, when these studies were identified, we contacted authors and data for patients with TKR were provided for two studies.[45 51] The primary outcome of interest in this review was pain at 6 months or longer after TKR, and therefore we did not include studies that used a composite pain and function measure to assess outcome, for example the total Oxford Knee Score[58] or WOMAC.[52] This is because when such composite measures are reported without any separation of pain from function, it is not possible to use the scores to assess pain per se. Preoperative risk factors for postoperative pain and functional limitations are different,[18 59] and therefore it is important to assess pain and function as distinct outcomes. Separate pain and function scores can be calculated for the most commonly used patient-reported outcome measures, the WOMAC[60] and the Oxford Knee Score,[61] and future studies would benefit from analysing these outcomes separately. Research on postoperative risk factors is limited by heterogeneity in how and when risk factors and outcomes are assessed. If greater standardisation could be achieved, such as through the implementation of core outcome sets,[33] future systematic reviews may be able to pool data in meta-analysis to provide evidence for postoperative patient-related risk factors for chronic pain after TKR.

Much of the research evaluating risk factors for outcomes after TKR has focused on the preoperative period rather than the period after surgery.[12] Numerous preoperative patient-related factors and their association to chronic pain have been evaluated, including knee pain severity and duration, pain at other sites, comorbidities, function, depression, social support, anxiety, fear of movement, pessimism and quality of life.[12] In comparison, our review found that the current extent of research into postoperative risk factors is narrow, and

**Table 5** Studies evaluating postoperative psychological factors as risk factors for chronic pain after TKR

| Author and year | Number in analysis | Risk factor measurement | Outcome(s) | Univariable or multivariable analysis | Association | Results summary |
|---|---|---|---|---|---|---|
| Stephens et al, 2002[48] | 63 | Perceived positive social support (MOS social support survey) at 6 weeks | WOMAC pain at 6 months | Multivariable hierarchical multiple regression | Yes | Beta=−0.29, SE=0.09, p≤0.05 |
| | | Perceived negative social support (four items) at 6 weeks | | | Yes | Beta=−0.27, SE=0.14, p≤0.05 |
| | | Active coping (Vanderbilt Multidimensional Pain Coping Inventory Active Coping scale) at 6 weeks | | | No | Beta=−0.14, SE=0.01 |
| | | Avoidant coping (Vanderbilt Multidimensional Pain Coping Inventory Avoidant Coping scale) at 6 weeks | | | No | Beta=0.21, SE=0.01 |
| Edwards et al, 2009[39] | 43 | Catastrophising (Coping Strategies Questionnaire catastrophising subscale) at 1 month and 3 months | Global pain VAS at 6 and 12 months Night pain VAS at 6 and 12 months | Multivariable generalised estimating equation model | No Yes | Catastrophising at a previous time point was not a predictor of global pain at a future time point (estimate=2.1, SE=2.2, t=0.9, p=0.35). Catastrophising at a previous time point was a predictor of night-time pain at a future time point (estimate=5.1, SE=2.5, t=2.0, p=0.04). |
| | | Depression (Center for Epidemiological Studies Depression Scale at 1 month and 3 months) | Global pain VAS at 6 and 12 months Night pain VAS at 6 and 12 months | | Yes No | Depression at a previous time point was a predictor of global pain at a future time point (estimate=0.67, SE=0.30, t=2.2, p=0.03). Depression at a previous time point was not a predictor of night-time pain at a future time point (estimate=0.40, SE=0.33, t=1.2, p=0.24). |
| Pinto et al, 2013[45] | 42 | Anxiety scale (Hospital Anxiety and Depression Scale) at 48 hours | NRS at 4–6 months | Hierarchical logistic regression | Yes | Exp(B)=1.713 (95% CI 1.104 to 2.657), p=0.016 |
| Kocic et al, 2015[50] | 78 | Fear of movement (Tampa Scale of Kinesiophobia) at 2 weeks | NRS at 6 months | Univariable comparison of means | Yes | Patients with high fear of movement had more pain (mean=3.24, SD=1.98) than patients with low fear of movement (mean=1.81, SD=1.5), p=0.0035. |

NRS, Numerical Rating Scale; TKR, total knee replacement; VAS, Visual Analogue Scale; WOMAC, Western Ontario and McMaster Universities Osteoarthritis Index.

further research is needed. Searches of ClinicalTrials.gov found that a number of studies are ongoing in this field, suggesting the evidence base will continue to grow and develop. Assessing potential postoperative risk factors is important as some factors may be more associated with outcome when measured in the postoperative period, rather than in the preoperative period.[62] Prediction of chronic postsurgical pain has been found to be strongest when assessing both preoperative and postoperative risk factors.[20] Factors specific to the postoperative recovery period, such as acute postoperative pain, and factors that span the perioperative period, such as anxiety, have the potential to influence outcomes. Identification of both preoperative and postoperative risk factors could inform the development of comprehensive care packages to improve outcomes.

Despite the lack of sufficient evidence about postoperative risk factors, research has evaluated whether early postoperative interventions improve longer term outcomes after TKR. The long-term effects of pharmacological interventions to reduce pain severity in the early postoperative period have been evaluated, both in patients undergoing TKR and other surgical procedures.[21 22] While effective at reducing acute postoperative pain, numerous perioperative pharmacotherapies are not effective at preventing chronic postsurgical pain. Similarly, outpatient physiotherapy interventions to improve early postoperative function have little effect on long-term pain.[23 24] This may be because acute postoperative pain and functional limitations are not risk factors for chronic pain after TKR, or it may be that these interventions require evaluation in trials that are focused on high-risk patients. However, before evaluation of such stratified models of care is possible, more research is needed to identify postoperative patient-related risk factors for chronic pain after TKR.

In conclusion, this systematic review found insufficient evidence to draw conclusions about the association between any postoperative patient-related factor and chronic pain after TKR. To complement this research, systematic reviews are ongoing to evaluate the effectiveness of preoperative, perioperative and postoperative interventions in preventing chronic pain after TKR (PROSPERO reference CRD42017041382). Further high-quality research is required to provide robust evidence on postoperative risk factors, and inform the development and evaluation of targeted interventions to optimise patients' outcomes after TKR.

**Acknowledgements** We would like to thank all the study authors who took the time to reply to our requests for further clarification or additional data.

**Contributors** All authors contributed to the concept and design of the study. AB, JD and VW contributed to the acquisition and analysis of data. VW drafted the article, and AB, JD and RG-H revised it critically for important intellectual content. VW and AB take responsibility for the integrity of the work as a whole, from inception to finished article.

**Funding** This article presents independent research funded by the National Institute for Health Research (NIHR) under its Programme Grants for Applied Research (RP-PG-0613-20001). The views expressed are those of the authors and not necessarily those of the NHS, the NIHR or the Department of Health. The funder had no role in the study design, collection, analysis and interpretation of data; in the writing of the manuscript; or in the decision to submit the manuscript for publication.

**Competing interests** None declared.

**Provenance and peer review** Not commissioned; externally peer reviewed.

**Data sharing statement** No additional data are available.

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
