## [Reviewer comments · BMJ Open]

ARTICLE DETAILS

TITLE (PROVISIONAL)	Post-operative patient-related risk factors for chronic pain after total knee replacement: a systematic review
AUTHORS	Wylde, Vikki; Beswick, Andrew; Dennis, Jane; Gooberman-Hill, Rachael

VERSION 1 – REVIEW

REVIEWER	RJH Custers University Medical Centre Utrecht, The Netherlands
REVIEW RETURNED	03-Jul-2017

GENERAL COMMENTS	The authors have written a clear review on the topic of post-operative patient-related risk factors (pain, function and psychosocial factors) for chronic pain after TKR.
---

REVIEWER	Colin McCartney University of Ottawa, ON, Canada
REVIEW RETURNED	06-Jul-2017

GENERAL COMMENTS	Overall this is a relevant, timely and well written systematic review examining postoperative predictors of chronic pain after knee replacement. There are some minor typos in the manuscript and I wonder if the authors should have examined postoperative opioid consumption in addition to pain in terms of predicting chronic pain.
--

REVIEWER	Christophe Aveline Département d'anesthésie et de réanimation chirurgicale Hôpital Privé Sévigné 35517 Cesson Sévigné, France
REVIEW RETURNED	25-Jul-2017

GENERAL COMMENTS	The authors proposed a systematic review of predictive factors of chronic pain after total knee replacement (TKR) for osteoarthritis. The primary endpoint was pain measured at 6months or longer after surgery. One aim was to determine the relations between patients-related risk factors of pain in the in the first three weeks after TKR and outcome. Despite a rigorous research using specific criteria, no quantitative analysis could be performed given the various and heterogeneous methods used to evaluate predictive factors and outcome.
--

No comments on methodology except the lack of search of unpublished studies in the ClinicalTrial registry. The authors detailed the PRISMA 2009 checklist and search terms. The statistical section needs no comments as no quantitative analysis was performed.

It seems, however, reading data on the immediate postoperative analgesia, that one major target of treatment after TKR is this postoperative period. The long-term functional outcome of these patients seems to be part of a painful trajectory that will depend on the quality of the surgical realization, its indication and the quality of the postoperative rehabilitation for which analgesia is a crucial part. The link between acute and chronic pain is multifactorial and difficult to assay. After TKR, chronic pain can affect patients after an asymptomatic period and presents frequently different characteristics compared to pre et peroperative pain. This aspect of pain intensity is difficult to predict with preoperative clinical factors as preoperative pain or functional limitation. As such, a quantitative analysis of data from VAS scores and postoperative opioid consumption might be able to be interesting in this study to quantify the actual part of analgesia in the occurrence of chronic pain after TKR.

Minor remarks for discussion:

One study (ref 34 in the text) is a retrospective analysis and must be described as such in the table 1.

The study (ref 43) is a secondary analysis of APEX study previously published (Pain 2015; 156: 1161-70). This reference (43) attempted to model pre and peroperative clinical elements. The initial study was a prospective randomized study evaluating the impact on the prevalence of chronic pain after TKR and THR of general or spinal anesthesia combined to a single-shot femoral nerve block with or without periarticular infiltration. No significant differences occurred between the standard strategy and infiltration for TKR patients. This is important, as infiltration is one of the main procedures currently used for TKR in ERAS protocols and also used in another study proposed by authors (ref 45).

One study included emergent and scheduled patients (ref 46), TKR and osteosynthesis were included and not analyzed separately. Again, analgesia was not standardized. Some patients were treated by ketamine as antihyperalgesic drug and, despite a higher tendency of chronic pain in ketamine's patients; this was not included in multivariate analysis.

On example of non-standardized analgesia is the ref 47, which mixed epidural analgesia, nerve block, infiltration, ketamine, and gabapentinoid. All these procedures were not analyzed.

One study (ref 45) associated a femoral perineural catheter (maintained 4 days) and periarticular infiltration but systemic analgesia was not standardized. Some patients received gabapentinoid, frequently used in ERAS protocol as antihyperalgesic drugs, although no analysis has been done on this criterion.

A study evoked the role of preoperative opioid use as risk factor for chronic pain (ref 37). The role of preoperative opioid must be discussed, as other studies have evocated this phenomenon (Eur J Anaesthesiol. 2015; 32: 255-61; J Bone Joint Surg Am. 2011; 93: 1988-93)

REVIEWER	Hans-Peter van Jonbergen Department of orthopedics Deventer hospital PO Box 5001 7400 GC Deventer The Netherlands
REVIEW RETURNED	01-Aug-2017

GENERAL COMMENTS	1. General comments The authors have performed a systematic review evaluating post-operative patient-related risk factors for chronic pain following total knee replacement. As the authors emphasize in the introduction, persistent pain following TKR is not uncommon and the impact is considerable. In this respect the study is of marked interest. I compliment the authors for writing this thorough and well written systematic review. It brings to our attention the need to identify modifiable factors related to persistent pain. 2. Introduction The authors pose a clear and important research question. Although this question is not novel, no systematic review has yet addressed this issue. As an orthopedic surgeon, I was surprised to read that chronic post-surgical pain is defined as pain present at three months after surgery. Why not pain at 12 months, or even 2 years after TKR surgery? It is well known that maximum recovery following total knee replacement requires at least 6 to 12 months. You decided to include only studies that reported a patient-reported outcome measure at 6 months or longer. In the Results section you report that some of the included studies evaluated whether pain severity between 8 weeks and 3 months postoperative was associated with chronic pain assessed at 6 months. For me this is not chronic pain, but a slow recovery. I doubt that in these patients maximum recovery has been achieved. Can you explain why you decided on evaluating 6 months outcome? 3. Methods The design is appropriate to answer the research question. 4. Results On page 9 you mention that follow-up assessments in the included studies differed, since four studies assessed outcomes at 6 months, five at 12 months, and the remainder between 3-7 years post-operative. Are the findings of these studies similar? 5. Discussion Assumptions, source of bias, and limitations are adequately described. Do you have an explanation for the results? 6. Tables I personally think that reporting the number of patients with chronic pain in each of the included studies may be helpful in order to assess whether a statistically significant finding is substantial enough to be clinically important.
---

VERSION 1 – AUTHOR RESPONSE

REVIEWER 1: RJH CUSTERS

Comment:

The authors have written a clear review on the topic of post-operative patient-related risk factors (pain, function and psychosocial factors) for chronic pain after TKR. Overall this is a relevant, timely and well written systematic review examining postoperative predictors of chronic pain after knee replacement. There are some minor typos in the manuscript and I wonder if the authors should have examined postoperative opioid consumption in addition to pain in terms of predicting chronic pain.

Authors' response:

Thank you for your helpful comments on our manuscript. We have now corrected the minor typographic errors in the manuscript, please accept our apologies for these. We agree that post-operative opioid consumption is important in the context of chronic pain after TKR. Our systematic review is focussed on patient-related risk factors and therefore we did not include opioid consumption, which we viewed as an intervention, rather than a patient factor. However, we are currently conducting systematic reviews of randomised controlled trials to evaluate the effectiveness of interventions in preventing chronic pain after TKR. These three reviews were registered on PROSPERO on 17th January 2017 (Reference CRD42017041382) and are evaluating interventions in the pre-operative period, peri-operative period and post-operative period. Within these reviews, we will include RCTs that have evaluated the effectiveness of analgesic interventions in preventing chronic pain after TKR.

We have clarified that we did not include analgesic use in this review on page 5 by modifying the sentence "The focus of this review was on patient-related risk factors with the potential for modification or use in targeting care, and therefore studies which assessed clinical risk factors (e.g. length of stay, post-operative complications, or radiographic measurements) or analgesic use were excluded". In the discussion we have now included reference to our ongoing systematic reviews on page 15, by adding the sentence "To complement this research, systematic reviews are ongoing to evaluate the effectiveness of pre-operative, peri-operative and post-operative interventions in preventing chronic pain after TKR (PROSPERO reference CRD42017041382)".

REVIEWER 2: CHRISTOPHE AVELINE

Comment 1

The authors proposed a systematic review of predictive factors of chronic pain after total knee replacement (TKR) for osteoarthritis. The primary endpoint was pain measured at 6months or longer after surgery. One aim was to determine the relations between patients-related risk factors of pain in the in the first three weeks after TKR and outcome. Despite a rigorous research using specific criteria, no quantitative analysis could be performed given the various and heterogeneous methods used to evaluate predictive factors and outcome. No comments on methodology except the lack of search of unpublished studies in the ClinicalTrial registry. The authors detailed the PRISMA 2009 checklist and search terms. The statistical section needs no comments as no quantitative analysis was performed.

Authors' response

Thank you, we agree with Reviewer 2 that we should have attempted to summarise ongoing observational studies. Thus, we have undertaken a search of ClinicalTrials.gov and identified 5 ongoing observational studies which are summarised in Appendix 3. We have included details of these searches on page 6 and page 11 of the manuscript. We have also added a sentence to the discussion on page 14 to say "Searches of ClinicalTrials.gov found that a number of studies are ongoing in this field, suggesting the evidence-base will continue to grow and develop".

We are pleased to have made this change as it increases the usefulness of the review for researchers in our field.

Comment 2

It seems, however, reading data on the immediate postoperative analgesia, that one major target of treatment after TKR is this postoperative period. The long-term functional outcome of these patients seems to be part of a painful trajectory that will depend on the quality of the surgical realization, its indication and the quality of the postoperative rehabilitation for which analgesia is a crucial part. The link between acute and chronic pain is multifactorial and difficult to assay. After TKR, chronic pain can affect patients after an asymptomatic period and presents frequently different characteristics compared to pre et peroperative pain. This aspect of pain intensity is difficult to predict with preoperative clinical factors as preoperative pain or functional limitation. As such, a quantitative analysis of data from VAS scores and postoperative opioid consumption might be able to be interesting in this study to quantify the actual part of analgesia in the occurrence of chronic pain after TKR.

The study (ref 43) is a secondary analysis of APEX study previously published (Pain 2015; 156: 1161-70). This reference (43) attempted to model pre and peroperative clinical elements. The initial study was a prospective randomized study evaluating the impact on the prevalence of chronic pain after TKR and THR of general or spinal anesthesia combined to a single-shot femoral nerve block with or without periarticular infiltration. No significant differences occurred between the standard strategy and infiltration for TKR patients. This is important, as infiltration is one of the main procedures currently used for TKR in ERAS protocols and also used in another study proposed by authors (ref 45).

One study included emergent and scheduled patients (ref 46), TKR and osteosynthesis were included and not analyzed separately. Again, analgesia was not standardized. Some patients were treated by ketamine as antihyperalgesic drug and, despite a higher tendency of chronic pain in ketamine's patients; this was not included in multivariate analysis.

On example of non-standardized analgesia is the ref 47, which mixed epidural analgesia, nerve block, infiltration, ketamine, and gabapentinoid. All these procedures were not analyzed.

One study (ref 45) associated a femoral perineural catheter (maintained 4 days) and periarticular infiltration but systemic analgesia was not standardized. Some patients received gabapentinoid, frequently used in ERAS protocol as antihyperalgesic drugs, although no analysis has been done on this criterion.

A study evoked the role of preoperative opioid use as risk factor for chronic pain (ref 37). The role of preoperative opioid must be discussed, as other studies have evocated this phenomenon (Eur J Anaesthesiol. 2015; 32: 255-61; J Bone Joint Surg Am. 2011; 93: 1988-93)

Authors' response

Thank you for highlighting this point, we agree that it is very important to consider anaesthetic regimes and analgesia use in studies of chronic pain after TKR. As we explain in response to the comments of Reviewer 1, we are focussed on patient-related risk factors in this review, rather than interventions to reduce pain. We wish to let the reviewer know that in other systematic reviews that we are currently conducting we are evaluating anaesthetic interventions and analgesia. In particular, our other reviews evaluate the effectiveness of pre-operative, peri-operative and post-operative interventions in preventing chronic pain after TKR (PROSPERO reference CRD42017041382). We have now clarified that we did not include analgesic use in this review on page 5. In the discussion we have now included reference to our ongoing systematic reviews on page 15. Your thoughtful comments on this topic have reiterated to us the importance of conducting these reviews, which we very much appreciate.

Comment 3

One study (ref 34 in the text) is a retrospective analysis and must be described as such in the table 1.

Authors' response

Thank you. We have now amended the manuscript to clarify that reference 34 was a retrospective analysis of a RCT (page 9 and Table 1).

REVIEWER 3: HANS-PETER VAN JONBERGEN

Comment 1

General: The authors have performed a systematic review evaluating post-operative patient-related risk factors for chronic pain following total knee replacement. As the authors emphasize in the introduction, persistent pain following TKR is not uncommon and the impact is considerable. In this respect the study is of marked interest. I compliment the authors for writing this thorough and well written systematic review. It brings to our attention the need to identify modifiable factors related to persistent pain.

Authors' response

We would like to thank Mr Van Jonbergen for his positive comments.

Comment 2

Introduction: The authors pose a clear and important research question. Although this question is not novel, no systematic review has yet addressed this issue. As an orthopedic surgeon, I was surprised to read that chronic post-surgical pain is defined as pain present at three months after surgery. Why not pain at 12 months, or even 2 years after TKR surgery? It is well known that maximum recovery following total knee replacement requires at least 6 to 12 months. You decided to include only studies that reported a patient-reported outcome measure at 6 months or longer. In the Results section you report that some of the included studies evaluated whether pain severity between 8 weeks and 3 months postoperative was associated with chronic pain assessed at 6 months. For me this is not chronic pain, but a slow recovery. I doubt that in these patients maximum recovery has been achieved. Can you explain why you decided on evaluating 6 months outcome?

Authors' response

Thank you for raising this important point. We define chronic pain as pain that is present at 6 months after TKR surgery based on the research literature on recovery trajectories. Although chronic post-surgical pain is defined as pain that is present at ≥ 3 months after surgery (Werner et al, 2014), the research literature has shown that recovery takes longer than this after TKR surgery, with most of the improvement in pain occurring in the first 6 months (Lenguerrand et al 2016, Halket et al 2010, Naylor et al 2009, Davis et al, 2011). There is only a slight improvement in pain between 6 months and 12 months and then outcomes plateau after 12 months (Wylde et al, in press). Therefore, it was most appropriate to include studies which assessed pain at 6 months after surgery. We have provided an explanation of our rationale for this on page 7, which states: "Chronic post-surgical pain is defined as pain present at three months after surgery [7], however research has shown that most of the improvement in pain occurs in the first 3-6 months after TKR surgery [29-31]. Therefore, six months post-operative was deemed an appropriate time point to assess chronic pain."

References

Davis AM, Perruccio AV, Ibrahim S, Hogg-Johnson S, Wong R, Streiner DL, et al. The trajectory of recovery and the inter-relationships of symptoms, activity and participation in the first year following total hip and knee replacement. *Osteoarthritis and Cartilage*. 2011; 19(12):1413–21

Halket A, Stratford PW, Kennedy DM, Woodhouse LJ. Using hierarchical linear modeling to explore predictors of pain after total hip and knee arthroplasty as a consequence of osteoarthritis. *The Journal of arthroplasty*. 2010; 25(2):254–62

Lenguerrand E, Wylde V, Gooberman-Hill R, Sayers A, Brunton L, Beswick A, Dieppe P, Blom AW. Trajectories of pain and function after primary hip and knee arthroplasty: the ADAPT cohort study. PLOS ONE, 2016, 12;11(2):e0149306.

Naylor JM, Harmer AR, Heard RC, Harris IA. Patterns of recovery following knee and hip replacement in an Australian cohort. Australian health review: a publication of the Australian Hospital Association. 2009; 33(1):124–35

Werner MU, Kongsgaard UE. I. Defining persistent post-surgical pain: is an update required? Br J Anaesth 2014; 113: 1-4.

Wylde V, Dixon S, Miller L, Whitehouse M, Blow AW. 5 Year Patient Reported Outcomes and Survivorship of the Triathlon Total Knee Replacement: A Cohort Study. Acta Orthopaedica Belgica, in press

Comment 3

Methods: The design is appropriate to answer the research question.

Authors' response

Thank you.

Comment 4

Results: On page 9 you mention that follow-up assessments in the included studies differed, since four studies assessed outcomes at 6 months, five at 12 months, and the remainder between 3-7 years post-operative. Are the findings of these studies similar?

Authors' response

As discussed in response to comment 2, research has shown that there is little improvement in pain after 6 months post-operative. In light of this, we considered all studies together, regardless of their timing of outcomes assessment. On re-reviewing the findings, there appears to be no apparent pattern in findings based on when outcomes were assessed, likely due to the fact that there is considerable heterogeneity in the assessment of risk factors and outcomes. To address this in the manuscript, we have modified a sentence on page 14 to read "Research on post-operative risk factors is limited by heterogeneity in how and when risk factors and outcomes are assessed."

Comment 5

Discussion: Assumptions, source of bias, and limitations are adequately described. Do you have an explanation for the results?

Authors' response

The finding from this systematic review is that there is insufficient evidence to draw any firm conclusions on post-operative patient-related risk factors for chronic pain after TKR. This is because only a limited amount of research has been conducted on this topic, and what research has been done is limited by heterogeneity in how and when risk factors and outcomes are assessed. This is described in the discussion on page 13. The recommendation from this review is that greater standardisation is needed to allow future systematic reviews to conduct meta-analysis to provide evidence for post-operative patient-related risk factors for chronic pain after TKR.

Comment 6

Tables: I personally think that reporting the number of patients with chronic pain in each of the included studies may be helpful in order to assess whether a statistically significant finding is substantial enough to be clinically important.

Authors' response

We agree that this would be useful additional information to include in the review to aid the interpretation of findings. Unfortunately the majority of studies included in the review present pain outcome data as means and standard deviations, rather than the number of patients with chronic pain. Therefore we were unable to include this information.

VERSION 2 – REVIEW

REVIEWER	Christophe Aveline Hôpital Privé Sévigné 8 rue du Chêne germain 35517 Cesson Sévigné France
REVIEW RETURNED	28-Aug-2017
GENERAL COMMENTS	I have no further comments to make. The authors have modified their text in accordance with the proposals of the reviewers.